# Continual Adaptation of Vision Transformers for Federated Learning

**Shaunak Halbe, James Seale Smith, Junjiao Tian, Zsolt Kira**
*Georgia Institute of Technology*
*Correspondence: shalbe9@gatech.edu*

**Reviewed on OpenReview:** *https://openreview.net/forum?id=vsZ5A3Zxyr*

## Abstract

In this paper, we focus on the important yet understudied problem of Continual Federated Learning (CFL), where a server communicates with a set of clients to incrementally learn new concepts over time without sharing or storing any data. The complexity of this problem is compounded by challenges from both the Continual and Federated Learning perspectives. Specifically, models trained in a CFL setup suffer from catastrophic forgetting which is exacerbated by data heterogeneity across clients. Existing attempts at this problem tend to impose large overheads on clients and communication channels or require access to stored data which renders them unsuitable for real-world use due to privacy. In this paper, we attempt to tackle forgetting and heterogeneity while minimizing overhead costs and without requiring access to any stored data. We study this problem in the context of Vision Transformers and explore parameter-efficient approaches to adapt to dynamic distributions while minimizing forgetting. We achieve this by leveraging a prompting based approach (such that only prompts and classifier heads have to be communicated) and proposing a novel and lightweight generation and distillation scheme to consolidate client models at the server. We formulate this problem for image classification and establish strong baselines for comparison, conduct experiments on CIFAR-100 as well as challenging, large-scale datasets like ImageNet-R and DomainNet. Our approach outperforms both existing methods and our own baselines by as much as 7% while significantly reducing communication and client-level computation costs. Code available at `https://github.com/shaunak27/hepco-fed`.

## 1 Introduction

Federated Learning (FL) is a privacy-preserving learning paradigm that enables learning a global model through communication with a distributed set of clients. These clients have exclusive access to private data, and collaborate with a central server to learn a shared task by communicating parameters such as model weights, gradients, or learning statistics. For example, the popular FedAvg McMahan et al. (2023) method works by iteratively aggregating client models by averaging their model weights. Classical FL methods such as FedAvg have garnered significant attention due to the increasing demand for user privacy and the growth of edge computing.

However, currently most federated learning methods focus on learning statically, that is across a fixed set of categories determined *a-priori*. In non-federated works, on the other hand, there has been a great deal of progress on learning an increasing number of categories incrementally, referred to as *continual* learning (and more specifically *class-incremental learning*) Hsu et al. (2018); van de Ven & Tolias (2019). In addition to the problem of catastrophic forgetting, incremental learning introduces challenges to typical federated learning (FL) scenarios by inherently involving non-Independent and Identically Distributed (non-IID) data, which has been shown to cause issues of model divergence Zhao et al. (2018); Li et al. (2020b). While *heterogeneous federated learning* Li et al. (2020b) approaches have been developed, they do not support the *dynamic data distributions* that occur in continual learning and the real-world. For example, such a setting has immense

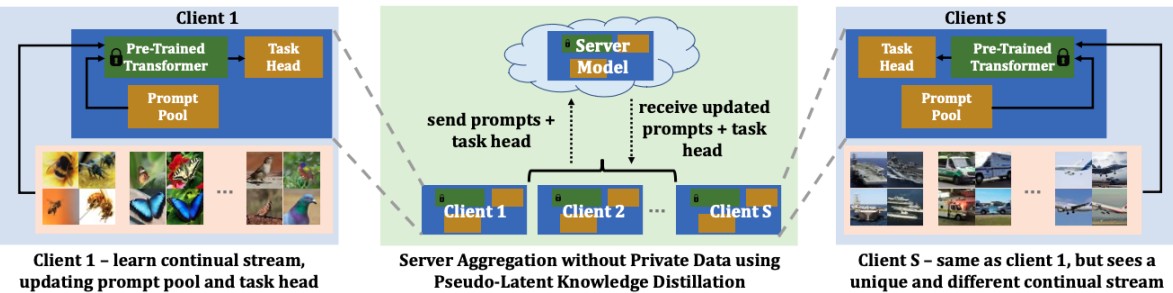

Figure 1: In Continual Federated Learning (CFL), clients learn from unique, continual data. We propose a prompt-based CFL approach, paired with a lightweight generation and distillation scheme, to consolidate client models at the server in a communication-efficient manner.

practical impact to applications such as healthcare, autonomous vehicles, and chat-bots Rieke et al. (2020); Nguyen et al. (2022).

Therefore, in this paper we look at the understudied problem of *Continual Federated Learning (CFL)* Yoon et al. (2021); Ma et al. (2022); Qi et al. (2023). While a few CFL methods exist, they address the issue of forgetting using approaches such as: reducing inter-client interference, knowledge distillation, and image-level generative replay Yoon et al. (2021); Dong et al. (2022); Zhang et al. (2023). Specifically, they often communicate full model weights, real/synthesized image-level data, or gradients. Additionally, some methods store old data in memory buffers or train a generative model to mimic local data; at the very least, all methods share complete models parameters with the server which can lead to privacy leaks with advancements in model inversion and other extraction techniques Carlini et al. (2023). *As a result, many of these methods fail to effectively uphold the principles of CFL, such as communication efficiency, computational efficiency, and privacy.*

To mitigate forgetting while adhering to the core principles of CFL, we propose HePCo: **H**eterogeneous **P**rompt **Co**nsolidation (Fig. 1). Our method is driven by the goals of (i) minimizing communication costs, (ii) improving client privacy, and (iii) client-level computation efficiency. We first propose to leverage *prompting*-based methods, which have shown successful results in the rehearsal-free continual learning setting Wang et al. (2022c;b). This also has the benefit of utilizing frozen Vision Transformer backbones, meaning that only prompts and classifiers have to be transmitted, reducing communication. The key contribution of our approach is then to answer the question of how to merge prompts from different clients in a scalable manner. Towards this end, we propose a lightweight method for generating pseudo-data *in the latent space* and distilling client model information. Importantly, we distill data from both the past task label distribution as well as the current task label distribution, preventing both catastrophic forgetting and performance degradation due to client heterogeneity. *In summary, we make the following key contributions:*

1. We modify and extend popular continual prompting based approaches to the setting of Continual Federated Learning (CFL), implementing a range of methods that we will open-source.

2. We introduce a light-weight pseudo-latent knowledge distillation mechanism to aggregate client models trained on non-IID data partitions. Importantly, our approach incurs no additional client-side computation and does not store or generate training data.

3. We outperform both existing methods and contributed baselines by as much as 7% while drastically reducing communication costs by only sharing a small set of model parameters, which constitutes only ~9.5% of the total parameters.

## 2 Related Work

**Prompting for Continual Learning.** Continual learning algorithms fall into several primary groups. Some methods involve expanding the model's architecture as new tasks arise Ebrahimi et al. (2020); Lee et al. (2020); Lomonaco & Maltoni (2017); Maltoni & Lomonaco (2019); Rusu et al. (2016), while others regularize the model with prior task knowledge either in the weight space or the prediction space Ahn et al. (2021); Aljundi et al. (2018); Hou et al. (2018); Kirkpatrick et al. (2017); Li & Hoiem (2017); Zenke et al.

(2017). Additionally, rehearsal-based methods leverage stored data or samples from a generative model Bang et al. (2021); Chaudhry et al. (2019b); Hayes et al. (2019); Hou et al. (2019); Lopez-Paz & Ranzato (2017); Ostapenko et al. (2019); Rebuffi et al. (2017b); Shin et al. (2017); von Oswald et al. (2019); van de Ven et al. (2020). Despite their efficacy, these methods might compromise data privacy and impose substantial memory costs, which justifies the need for *rehearsal-free strategies*. In the realm of rehearsal-free continual learning, some studies focus on an online learning perspective utilizing a pre-trained model Hayes & Kanan (2019); Lomonaco et al. (2020), or explore prototype-based techniques to avert catastrophic forgetting Yu et al. (2020); Wu et al. (2021); Zhu et al. (2021). Other work proposes deep-model inversion to produce images for rehearsal, yet the computational expense and data-privacy issues render this approach challenging Choi et al. (2021); Kaissis et al. (2021); Smith et al. (2021); Yin et al. (2020). NCDwF Joseph et al. (2022) performs classifier inversion and pseudo-replay in the latent-space to reduce forgetting in a novel class-discovery setting. Recent works have demonstrated that **prompting** within a static, pre-trained transformer model for continual learning achieves state of the art performance without any rehearsal data Smith et al. (2022); Wang et al. (2022d;a). Our work builds on the foundations of these prompting methods, but in a federated setting.

**Heterogeneous Federated Learning.** Federated learning involves a number of clients learning over local data, to be aggregated by a global server. One of the earliest methods for this is FedAvg McMahan et al. (2023), which simply averages the parameters of the clients weighted by the relative amount of data they were trained on. The most investigated challenge in federated learning (FL) is client heterogeneity, which can cause local models to diverge and the aggregated global model to perform sub-optimally Li et al. (2020b). In this work, we discuss FL works tackling this issue in two categories: parameter-averaging methods and knowledge distillation methods. To combat high heterogeneity in local data, methods like FedProx Li et al. (2020a), FedPD Zhang et al. (2020), FedDyn Acar et al. (2021), and SCAFFOLD Karimireddy et al. (2019) are used, out of which FedProx is a *stateless* algorithm useful in settings where the number of clients is very large which prevents the server from keeping a track of all participating clients. The rest are *stateful* algorithms that maintain client states. Knowledge-distillation-based methods Mora et al. (2022); Lin et al. (2020); Sattler et al. (2021) usually use additional data to perform distillation on the server or client side to robustly aggregate local models. FedFTG Zhang et al. (2022a) operates in a data-free setting by generating pseudo-data through inversion in the image space. In contrast, we achieve knowledge distillation using pseudo-data generated in latent space to 1) reduce computation overhead stemming from inversion in a higher-dimensional space and 2) fine-tune both the classifier and the prompt components simultaneously which is essential to mitigate forgetting.

**Continual Federated Learning.** Most of the current CFL methods suffer from various limitations in terms of performance, efficiency and privacy. FedWeiT Yoon et al. (2021) aims to learn better client models by leveraging indirect experience of other client models. The objective introduced in this work minimizes interference across client weights. A knowledge base of previous task parameters for all seen clients is maintained at both the server and client side and is updated during each round through client-server communication. Each client selectively utilizes the parameters in this knowledge base through attention masks to gain indirect experience. FedWeiT incurs considerable overheads in terms of communication, computation and storage stemming from maintaining and updating this knowledge base. GLFC Dong et al. (2022) uses a prototype based approach with a memory buffer to store old data. This poses a threat to privacy of client data. CFed Ma et al. (2022) proposes a distillation based approach that makes use of an unlabelled dataset to aggregate client models as well as to rehearse old tasks. CFeD performs distillation at both client and server side. However the requirement for a curated dataset can severely impact real-world applicability. TARGET Zhang et al. (2023) attempts to combat forgetting through replay of old tasks driven by generated images. FedCIL Qi et al. (2023) leverages an Auxiliary Classifier GAN (ACGAN) to alleviate forgetting by synthesizing old images for replay. However, generating images to mimic local datasets can be viewed as a privacy risk especially in light of recent research on model inversion attacks Carlini et al. (2023). In contrast, our approach prioritizes client privacy by generating in the latent space, thereby eliminating the need to generate in the image space. Further, this benefits communication and compute efficiency. GAL Wang et al. (2023) introduces the task of Federated Continual Novel Class Learning, where FL methods are expected to discover and learn unlabelled novel class data. In contrast, our work explores a supervised continual learning setup where new labeled instances are incrementally introduced. The recent work Fed-CPrompt Bagwe et al. (2023), investigates the use of prompting for federated class-incremental learning. To combat client divergence,

their approach introduces a contrastive loss at the client side. Client aggregation is performed at the server using the standard FedAvg algorithm. Our work differs in that we do not introduce additional objectives at the client side. Instead, we propose a novel method for combining client models at the server to mitigate forgetting.

## 3 Problem Formulation

In this section, we describe the formulation by introducing the class-incremental learning and heterogeneous federated learning aspects in our CFL setting.

**Class-Incremental Federated Learning.** We focus on the *class-incremental* learning scenario, where a model is tasked with learning new classes over time. Under this setting, a global model is learned through a sequence of $N$ *global* tasks $T = \{T^1, T^2, ..., T^N\}$. Following the standard assumption in continual learning, the sets of categories[1] seen across distinct *global* tasks are mutually exclusive. As this is done in a federated setup, each task is learned through $R$ independent rounds by randomly sampling a set of *stateless* clients $C = \{c_1, c_2, c_3, ..., c_S\}$ in each round. In a *stateless* setting, the total number of clients is kept very large to simulate real-world FL applications (like mobile devices). The server does not keep track of clients as new clients are visited in each round. Further, previously seen data cannot be accessed at any time during the training.

**Heterogeneous Federated Learning.** To simulate a real-world heterogeneous federated learning scenario, we use three configuration parameters to control the level of heterogeneity with increasing granularity: *split ratio*, *category ratio*, and *imbalance ratio*. At the top level, the most common heterogeneity is varying local dataset sizes across the clients. A specific client $c_i$ can be exposed to a subset of the current task dataset $D^t$ as their local dataset $D_i^t$, and the size $|D_i^t|$ varies from each other. We denote this as the *split ratio* $\gamma = |D_i^t|/|D^t|$. At a lower level, the local dataset $D_i^t$ consists of a subset of the categories from those in the current global task. Specifically, a global task $T^t$ consists of categories $K^t$, where $|K^t|$ denotes the number of unique categories. In a given round $r$, each client $c_i$ sees data containing $K_i^t \in K^t$ categories. We denote $\kappa = |K_i^t|/|K^t|$ as the *category ratio* which is a value between 0 and 1. At the lowest level, each category can have a different amount of data. We follow Cao et al. (2019) to also create a long-tail distribution for local data which is different for each client. This distribution is governed by an *imbalance ratio* $\beta$. If $\beta = 1$, each client $c_i$ is allocated samples uniformly from $K_i^t$ categories. In summary, a smaller split ratio $\gamma$, a smaller category ratio $\kappa$, or a smaller imbalance ratio $\beta$ increases heterogeneity thereby increasing the complexity of the task. Formalizing the setting in this manner enables us to methodically vary the parameters, thereby simulating a heterogeneous setting. This formalization unifies the setups previously employed separately across federated and continual learning landscapes Li et al. (2020a); Rebuffi et al. (2017a) combining their distinct characteristics.

As discussed before, combining client models in a heterogeneous setting causes the obtained model to forget global knowledge from the previous rounds as shown by Lee et al. (2022); Hsu et al. (2019). Also, training clients on locally-available datasets induces forgetting of global knowledge outside the local distributions. *Intra-task forgetting* Ma et al. (2022) measures such performance drops induced by the data heterogeneity (non-IIDness) across clients. *Inter-task forgetting* measures the drop in performance on old tasks $\{T^1, T^2, ..., T^{t-1}\}$ after learning a new task $T^t$.

## 4 Background: L2P

L2P (Learning to Prompt) Wang et al. (2022d) is a continual learning method that learns a set of model embeddings (prompts) that can be dynamically inserted into a pretrained vision transformer. (ViT) Dosovitskiy et al. (2020). Prompts hold task-specific information that instructs the model to solve the corresponding tasks. L2P maintains a prompt pool $P = \{P_1, P_2, \cdots, P_M\}$ of size M, where $P_i \in \mathbb{R}^{L_p \times D}$ are prompt parameters with $L_p$ as the prompt length (chosen as a hyperparameter) and $D$ the embedding dimension. Each prompt

---

[1]We use the terms 'category', 'class', and 'label' interchangeably to refer to the target classification label.

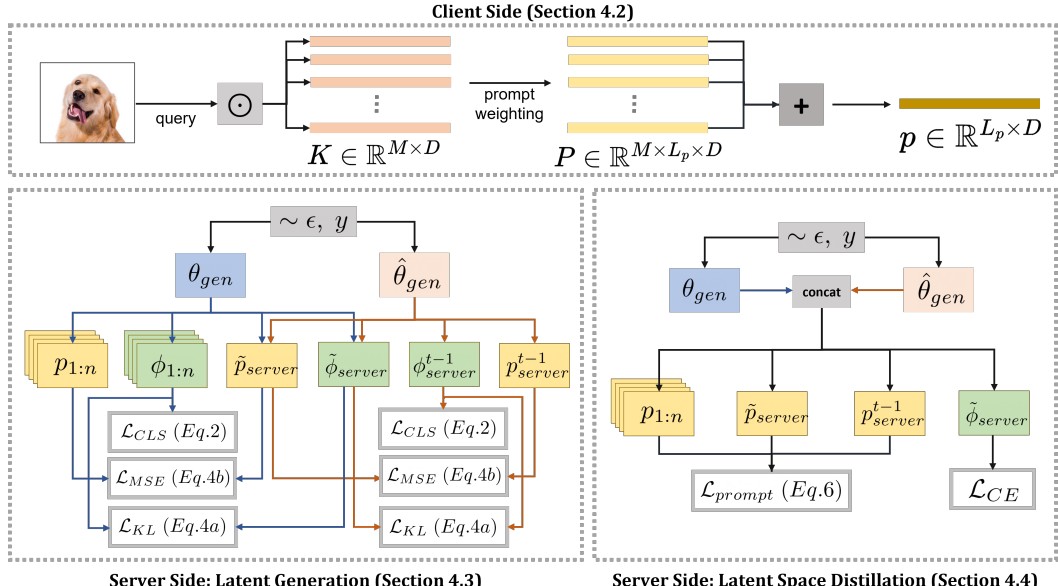

Figure 2: Latent generation and distillation with underlying *decomposed prompting* scheme.

$P_i$ has an associated key $k_i \in \mathbb{R}^{\mathbb{D}}$. An input image $x$ is converted into a visual query $q(x) \in \mathbb{R}^{\mathbb{D}}$ by passing it through the frozen vision transformer encoder $\theta_{pt}$. Prompts are selected from the pool by measuring the cosine similarity between associated keys and the visual query. Top $N$ prompts that have the highest key-query similarity are chosen to be inserted into the transformer. Here, the ViT encoder is frozen and the prompts and keys are learned using two separate optimization losses.

## 5    Method

In this section, we describe our novel approach called HePCo (**H**eterogenous **P**rompt **Co**nsolidation) which tackles forgetting and heterogeneity using a data-free distillation strategy applied in the model's latent space. Knowing the limitations of FedAvg under non-IID data partitioning, we propose to fine-tune the global model at server side by distilling knowledge from client models. However, unlike prior CFL works, we first propose to leverage the current state of art *prompting methods* in continual learning. Such methods optimize learnable parameters that augment the input to a transformer model (prompt tuning) or its underlying attention mechanism (prefix tuning). These methods have been shown to obtain strong performance in traditional rehearsal-free continual learning settings. These prompting-based approaches Wang et al. (2022d); Smith et al. (2022) can be thought of implicit mechanisms to isolate model parameters across tasks. Implementing a prompting scheme at the client level can thus prevent forgetting of past tasks while efficiently adapting to new tasks.

Despite these advantages, there is a key challenge in applying prompting to a federated setting: It is not obvious how the server should *combine* prompts learned by the individual clients on heterogeneous sources of data. Naively averaging the prompt weights is suboptimal (as we show in Sec. 6) and simply maintaining a growing prompt pool scales poorly with the number of clients. Our key novelty is therefore to propose a lightweight *distillation method*, applied to the latent-space of the model, which greatly mitigates intra-task and inter-task forgetting. Crucially, rather than averaging different client prompts, we perform distillation of these prompts in a data-free manner by *generating pseudo-data*. Importantly, the generation and distillation operations are computationally cheap as they are carried out in the latent space of the model. This design prioritizes privacy and efficiency, which are crucial for federated learning. In summary, with our method, communication costs between clients and the server are low, heterogeneous information is effectively aggregated, and the method achieves state-of-art performance. Below we detail our method and depict it in Fig. 2.

### 5.1 Client Side: Decomposed Prompting

While L2P is quite successful in protecting against forgetting, its performance is limited by certain design choices, as noted by Smith et al. (2022). We therefore adapt L2P to side-step these issues, including for our baselines, resulting in better accuracy across the board. Specifically, rather than using discrete prompts obtained via a hard maximum function which restricts capacity and introduces an additional hyperparameter ($N$, corresponding to top-$N$ prompts), we form our final prompt $p$ by taking sum of the $P_i$ weighted by their cosine scores. Such a prompt is inserted into a subset of the self attention layers of the ViT encoder. This subset of layers is determined by hand as in Smith et al. (2022).

In the federated learning setting, each client hosts the aforementioned prompting scheme and learns the key and prompt matrices along with the classifier in an end-to-end fashion while keeping the ViT backbone frozen. After learning the local task, in our method, the clients transfer the key, prompt and classifier weights to the server. By sending only a limited number of parameters to the server, the overall communication load is significantly reduced compared to sharing the entire model. Our approach requires clients to share only the prompt and classifier weights. The server lacks knowledge of the exact locations for inserting the prompts, as this information is known only to the respective clients. To reconstruct the exact client model, the server would need to conduct an exhaustive search over the number of layers in the transformer and try various combinations to determine the precise insertion positions. This method represents a positive step towards enhancing client privacy compared to approaches that share complete model weights.

### 5.2 Server Side: Latent Generation

At the end of each round, the server receives $|C|$ prompt and classifier weights collected from the active clients. Let $w_c$ indicate a weight parameter that encompasses key, prompt and classifier weights. In the first stage, we obtain the server model by averaging client weights as: $w = \frac{1}{C} \sum_{c \in C} w_c$. We call these as the *provisional server weights.*

Due to data heterogeneity, the local model weights diverge which degrades the performance of this aggregated model. We therefore propose to fine-tune the server model using data-free distillation in the latent space to prevent this degradation. We generate pseudo data in the latent space of the *visual query* $q(x) \in \mathbb{R}^{\mathbb{D}}$ which is essentially the output space of the vision encoder. The advantage of generating in this space is that it allows us to fine-tune *both* the classifier and the key-prompt weights *without needing a forward-pass through the encoder*. We use a lightweight feedforward neural network as our conditional generator with a $D$ dimensional output. This generator takes as input a class label (from categories in current task $t$) and a noise vector of dimension $D_{noise}$ sampled from the standard normal distribution $\mathcal{N}(\mathbf{0}, \mathbf{1})$. We encode the class label using an embedding layer and concatenate the obtained class embedding with the noise vector to form the input of the generator. From the generator, we obtain a pseudo latent of dimension $D$ conditioned on the class label as follows:

$$z = G(\epsilon, y; \theta_{gen}), \tag{1}$$

where $z \in \mathbb{R}^D$ is the generated pseudo latent and $\epsilon \in \mathbb{R}^{D_{noise}} \sim \mathcal{N}(\mathbf{0}, \mathbf{1})$ is the noise vector. For effective knowledge distillation, pseudo data should conform to the latent space of the client models. We optimize for a classification loss which is a weighted sum of classification losses for each individual client, similar to Zhang et al. (2022b). The total classification loss can be given as:

$$\mathcal{L}_{cls} = \sum_{c \in C} \mathcal{L}_{cls}^c \quad \text{and,} \tag{2}$$

$$\mathcal{L}_{cls}^c = \sum_{c \in C} \mathcal{L}_{CE}(\phi(z; w_c), y) \tag{3}$$

where $\mathcal{L}_{cls}^c$ is the cross-entropy loss between the prediction of local model $c$ given latent $z$ and sampled class label $y$. Here, $\phi$ denotes the classifer (last layer). However, optimizing for just the classification loss encourages the generator to produce pseudo latents which are easy to be classified and hence less effective for distillation. Our goal is to generate latents that create a discrepancy between the provisional server model and clients, thereby providing a learning opportunity for the server. To promote the generation of such *hard samples*, we

maximize two disagreement losses (one for prompts and one for classifier) between server and client models. As stated earlier, latents in this space can be forwarded through both prompting and classifier mechanisms. Hence, to measure the disagreement with respect to the classifier, we compute the Kullback-Leibler (KL) divergence between the predictions of the classifier corresponding to the provisional server model and each individual client. Next, to measure the disagreement with respect to the prompting module, we introduce a Mean-Squared Error (MSE) loss between the final prompts generated by the server and all clients. By training the generator to maximize these losses, we increase the potency of pseudo-latents for distillation in both parts of the network.

$$\mathcal{L}_{KL} = \sum_{c \in C} \sigma(\phi(z; w)) || \sigma(\phi(z; w_c)) \quad \text{(4a)} \qquad\qquad \mathcal{L}_{MSE} = \sum_{c \in C} \mathcal{L}_{MSE}(\rho(z; w), \rho(z, w_c)) \quad \text{(4b)}$$

where $\sigma$ denotes the softmax function and $\rho$ denotes the prompting mechanism described in 5.1. We train the generator by optimizing for these losses jointly as:

$$\min_{\theta_{gen}} \mathbb{E}_{\epsilon \sim \mathcal{N}(\mathbf{0},\mathbf{1})} \left[ \mathcal{L}_{cls} - \lambda_{KL} \mathcal{L}_{KL} - \lambda_{MSE} \mathcal{L}_{MSE} \right] \tag{5}$$

The trained generator is then used to perform data-free knowledge distillation which helps combat intra-task forgetting and allows the server model to achieve a high accuracy on the global data distribution of the current task. However, as the generator is trained to generate pseudo-data corresponding to the current task distribution only, a model fine-tuned with this pseudo-data suffers from inter-task forgetting as shown in our ablation experiments in Section 6.2.1. To prevent this, we train a separate copy of the generator ($\hat{\theta}_{gen}$) to generate latents corresponding to the previously seen tasks. At the server side, we assume access to the key, prompt and classifier weights corresponding to the previous task's global model which is the fine-tuned global model after the $R^{th}$ round of the task $t-1$. Similar to above, we optimize for the classification and disagreement losses jointly. Here the classification loss $\mathcal{L}_{cls}$ is computed for the previous task server model and $\mathcal{L}_{KL}$ and $\mathcal{L}_{MSE}$ are computed between this and the provisional server model. We empirically show that using pseudo-data corresponding to past tasks for knowledge distillation helps mitigate inter-task forgetting to a great extent.

## 5.3 Server Side: Latent Space Knowledge Distillation

Once the generator is trained to generate pseudo-latents corresponding to the current and previous tasks, we use it to efficiently fine-tune the provisional server model $w$. We use the pseudo-latents obtained from the generator to fine-tune both the classifier head and key-prompt weights ($K$ and $P$) without requiring a forward pass through the full model. We use the key, prompt and classifier weights corresponding to the current round client models and the last-task server model to fine-tune the server model. As it operates in a low dimensional latent space and updates a small subset of parameters, this distillation process is much cheaper in terms of computation compared to training the entire model. Also, this design does not require clients to share entire models with the server which reduces the client-server communication costs to a great extent and improves privacy of the client model. While we introduce additional server overhead, it is important to note that, in the context of CFL, clients are typically edge devices with limited computing power, while servers have ample computational resources. We prioritize client-level efficiency while making efficient use of the server's resources. In Section 6.2.2, we compute this overhead and show that it is competitive to that incurred by existing state-of-the-art (SOTA) methods.

To perform knowledge distillation, we first generate a batch of pseudo-data from the generators corresponding to the current round and previous task. We mix the current and previous task batches to form a single composite batch according to a hyperparameter named *replay ratio* which determines the size of the previous task batch relative to the current round batch. We use this composite batch of pseudo-latents to fine-tune the key-prompt weights and the classifier weights separately.

To fine-tune the key-prompt weights, we first obtain distillation targets in the form of final prompts $\boldsymbol{p}$ by passing the pseudo latents through the prompting mechanism of the teacher models (clients and previous-task server model). Notably, we do not require full models to generate these targets, as having only the key-prompt and classifier weights is sufficient. Now, to fine-tune the key-prompt weights of server model, we optimize

for the Mean Squared Error (MSE) loss between the final prompts predicted by the provisional server model and each individual teacher model (clients and previous-task server).

$$\mathcal{L}_{prompt} = \sum_{c \in C} \mathcal{L}_{MSE}^c + \zeta_{t-1}^y L_{MSE}^{t-1}, \tag{6}$$

where $\mathcal{L}_{MSE}^c$ denotes the MSE loss between client $c$ and the provisional server model and $L_{MSE}^{t-1}$ denotes the MSE loss between the provisional model and the previous task server model. Further, $\zeta_{t-1}^y$ is an indicator variable which is set to 1 if $y$ was seen in previous tasks and 0 if present in current task.

Next, we fine-tune the classifier layer of the provisional server model. As discussed in Section 5.2, pseudo-latents were obtained from the generator by conditioning on randomly sampled class labels. The composite batch of pseudo-latents used in the previous step is employed here again as input to the classifier, and the cross-entropy loss is computed between the predictions of the provisional server and the class labels upon which the pseudo-latents were conditioned. This approach allows us to fine-tune the classifier weights using the same batch of pseudo-latents used to fine-tune the key-prompt weights. Operating in the latent space allows us to efficiently fine-tune the key-prompt and classifier modules without requiring forward passes through the entire model. Our ablation studies in Section 6.2.1 highlight the efficacy of this approach.

## 6 Experiments

**Model Architecture.** We use the ViT-B/16 backbone Dosovitskiy et al. (2020) pretrained on Imagenet-1K Russakovsky et al. (2015) as the encoder for our method and all baselines. We use a prompt pool size ($M$) of 100 and a prompt length ($L_p$) of 20 with dimension ($D$) being 768 and insert prompts into 1-5 Multi-head Self Attention (MSA) layers of the ViT encoder following the standard practice Wang et al. (2022b) and perform prefix-tuning as done in Smith et al. (2022), by prepending prompts to the keys and values of the MSA layers. The classifier ($\phi$) is a fully-connected layer with input dimension $D$ and output dimension equal to the number of classes. We implement the generator $\theta_{gen}$ using a three layer fully-connected network and train it for 100 epochs. We encode the class label using an embedding matrix and concatenate the class embedding to the sampled noise vector before feeding into the generator.

**Datasets.** We conduct our experiments on three image classification datasets. First, we adapt CIFAR-100 Krizhevsky et al. (2009) to our formulation as it is a commonly used benchmark in CFL. Additionally, we evaluate our methods on the larger-scale ImageNet-R Hendrycks et al. (2021) and DomainNet Peng et al. (2019) which have been used in recent continual learning works Smith et al. (2022) but haven't been explored in a continual federated learning setting. These datasets capture real-world distribution shifts that can be challenging for models pre-trained on ImageNet to generalize to. The total number of classes for CIFAR-100, ImageNet-R and DomainNet are 100, 200 and 345 respectively. We divide these datasets into 10-task (CIFAR-100, ImageNet-R) and 5-task (DomainNet) benchmarks. The 10-task setups contain a longer task sequence with small number of classes per task whereas the 5-task setup has a shorter task sequence with more classes per task. CIFAR and ImageNet have 10 and 20 classes, while DomainNet has 69 classes per task. Following Wang et al. (2022a), we use 20% of the training set as our validation dataset to determine hyperparameters for our approach and all competing baselines.

**Configuration.** We learn each task through $R = 10$ communication rounds by selecting $C = 5$ stateless clients per round. Thus, we have 100 total rounds for a 10-task setup and 50 for a 5-task setup. For all experiments reported in Tables 1-3, we use a category ratio $\kappa = 0.6$ which means that if a task contains 10 categories, each active client is randomly assigned 6 of these categories. We analyze the affect of different category ratios in Sec. 6.2.1. Overall, HePCo outperforms existing methods across all category ratios, particularly excelling in scenarios with smaller category ratios, which signify higher heterogeneity and, consequently, a higher level of task complexity. Further, we use a split ratio $\gamma = 0.1$ which allows a client to be assigned 10% of the images corresponding to the subset of categories. We train local models for 10 epochs per round.

*We include additional implementation details in Appendix B.*

**Metrics.** We evaluate all methods using the standard continual learning metrics of (1) final average accuracy $A_N$ which is the accuracy averaged over all $N$ tasks after learning the $N^{th}$ task and (2) average forgetting

Table 1: **Results (%)** for the class-balanced setups reported over 3 independent trials. $A_N$ gives the accuracy averaged over tasks and $F_N$ gives the average forgetting.

| Datasets ($\beta = 1$) | CIFAR-100 | | ImageNet-R | | DomainNet | |
|---|---|---|---|---|---|---|
| Method | $A_N$ (↑) | $F_N$ (↓) | $A_N$ (↑) | $F_N$ (↓) | $A_N$ (↑) | $F_N$ (↓) |
| Prompting (Centralized) | 85.35 | - | 72.28 | - | 71.33 | - |
| FedAvg-FT | $10.23 \pm 1.10$ | $31.74 \pm 0.80$ | $12.03 \pm 0.75$ | $29.07 \pm 0.66$ | $18.76 \pm 0.44$ | $32.81 \pm 1.22$ |
| FedLwF.MC Rebuffi et al. (2017a) | $59.08 \pm 1.06$ | $12.39 \pm 0.76$ | $52.87 \pm 0.61$ | $13.34 \pm 0.38$ | $62.39 \pm 1.12$ | $10.76 \pm 0.50$ |
| FedAvg-Prompt | $67.34 \pm 1.42$ | $8.38 \pm 0.42$ | $51.15 \pm 0.68$ | $8.84 \pm 0.52$ | $51.03 \pm 2.23$ | $12.03 \pm 0.45$ |
| Fed-CPrompt Bagwe et al. (2023) | $69.38 \pm 1.39$ | $7.18 \pm 0.65$ | $52.24 \pm 0.66$ | $8.21 \pm 0.56$ | $60.38 \pm 0.78$ | $8.22 \pm 0.46$ |
| CFed Ma et al. (2022) | $72.26 \pm 1.56$ | $8.82 \pm 0.64$ | $45.64 \pm 1.32$ | $11.74 \pm 1.22$ | $63.32 \pm 0.78$ | $\underline{7.12 \pm 0.66}$ |
| TARGET Zhang et al. (2023) | $73.56 \pm 1.42$ | $\underline{6.83 \pm 0.91}$ | $52.38 \pm 1.16$ | $8.88 \pm 0.96$ | $61.84 \pm 1.66$ | $7.94 \pm 0.52$ |
| **HePCo (Ours)** | $\mathbf{76.54 \pm 1.14}$ | $\mathbf{6.61 \pm 0.73}$ | $\mathbf{59.96 \pm 0.94}$ | $\mathbf{7.08 \pm 0.40}$ | $\mathbf{64.01 \pm 0.36}$ | $\mathbf{6.83 \pm 0.31}$ |

Table 2: **Results (%)** for class-imbalanced setup reported over 3 independent trials. $A_N$ gives the accuracy averaged over tasks.

| Datasets | CIFAR-100 | | ImageNet-R | | DomainNet | |
|---|---|---|---|---|---|---|
| Method | $A_N$ (↑) | | $A_N$ (↑) | | $A_N$ (↑) | |
| Imbalance ratio ($\beta$) | $\beta = 0.05$ | $\beta = 0.01$ | $\beta = 0.05$ | $\beta = 0.01$ | $\beta = 0.05$ | $\beta = 0.01$ |
| FedAvg-FT | $8.81 \pm 1.53$ | $9.18 \pm 1.26$ | $9.26 \pm 1.02$ | $8.88 \pm 1.24$ | $13.02 \pm 1.29$ | $11.65 \pm 1.84$ |
| FedLwF.MC Rebuffi et al. (2017a) | $50.40 \pm 0.88$ | $40.39 \pm 1.06$ | $19.94 \pm 0.78$ | $13.34 \pm 1.41$ | $57.34 \pm 0.84$ | $52.46 \pm 0.72$ |
| FedAvg-Prompt | $62.72 \pm 1.79$ | $54.43 \pm 1.57$ | $36.51 \pm 0.86$ | $28.16 \pm 1.12$ | $47.73 \pm 1.25$ | $43.23 \pm 1.03$ |
| Fed-CPrompt Bagwe et al. (2023) | $65.42 \pm 1.12$ | $56.85 \pm 1.79$ | $39.03 \pm 1.04$ | $30.14 \pm 1.16$ | $54.44 \pm 0.80$ | $49.96 \pm 0.74$ |
| CFed Ma et al. (2022) | $\underline{70.26 \pm 1.20}$ | $\mathbf{62.04 \pm 1.62}$ | $34.62 \pm 1.41$ | $25.74 \pm 1.08$ | $\underline{59.89 \pm 0.68}$ | $55.22 \pm 0.80$ |
| TARGET Zhang et al. (2023) | $66.47 \pm 1.22$ | $58.13 \pm 1.54$ | $30.20 \pm 1.35$ | $19.84 \pm 1.41$ | $56.44 \pm 0.45$ | $51.82 \pm 0.58$ |
| **HePCo (Ours)** | $\mathbf{70.34 \pm 1.08}$ | $\underline{61.70 \pm 1.48}$ | $\mathbf{45.45 \pm 0.98}$ | $\mathbf{41.68 \pm 1.44}$ | $\mathbf{61.10 \pm 0.76}$ | $\mathbf{58.82 \pm 0.84}$ |

Chaudhry et al. (2019a); Lopes et al. (2017) $F_N$ which measures the drop in performance on previous tasks after learning a new task averaged over all $N$ tasks. As noted by Smith et al. (2022), $A_N$ is the more informative metric as it encompasses both forgetting and plasticity (new task performance). Our approach provides parameter efficiency which is reflected by reduced communication and local computation costs; albeit at the expense of a small overhead at the server. We quantify this communication efficiency by specifying the number of parameters shared by our approach relative to the original model size in 6.2.2.

**Baselines.** We compare our method against existing state-of-the-art approaches: TARGET Zhang et al. (2023), CFed Ma et al. (2022) and Fed-CPrompt as well as two strong baselines that we introduce. TARGET and CFeD mitigate forgetting by replaying old task data through generative methods or by assuming access to surrogate datasets. Fed-CPrompt uses a prompting-based approach similar to ours but introduces a contrastive loss at the client side. This additional loss aims to mitigate forgetting by alleviating heterogeneity across clients and tasks. At the server side, Fed-CPrompt aggregates clients using the standard FedAvg algorithm. In contrast, our approach does not introduce any additional computation at the client's end. Instead, we focus on altering the aggregation procedure at the server to combat forgetting. For fair comparison, we adapt Fed-CPrompt to our experimental setup and use the same Vision Transformer (ViT) architecture as in our method. Similarly for all other methods, we adapt their implementations to use the same ViT backbone with proper modifications. We tune hyperparameters for our proposed method and all compared baselines. We observe that the FedLwF method Ma et al. (2022) used in prior CFL work performs poorly owing to the complexity of our setting. As the heterogeneity across clients increases, coupled with an increase in the length of the global task sequence, the performance of FedLwF deteriorates catastrophically. We instead adapt LwF.MC with sigmoid binary cross-entropy loss as described in Smith et al. (2023) which is a strong continual learning baseline to our setting. We call this method FedLwF.MC which achieves a much better performance than its vanilla counterpart. Additionally, we introduce a simple yet strong prompting-based methods which we call FedAvg-Prompt. FedAvg-Prompt differs from our method in the model aggregation part at the server side where the clients are simply averaged to obtain the server model. Additionally for completeness, we report the performance of FedAvg-FT where the entire client models are sequentially finetuned on new task data (as opposed to learning only prompts and classifier) and aggregated using FedAvg. Finally, we report the performance of our *decomposed prompting* scheme in a centralized, traditional continual learning setting. This can be thought of as an upper bound performance for all prompt-based methods included here.

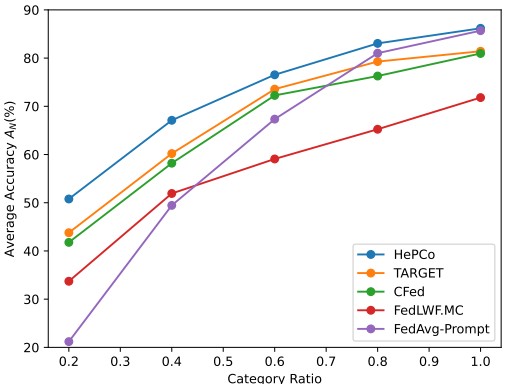

Figure 3: Comparison of the methods under different category ratios.

| Method | $A_N$ ($\uparrow$) | $F_N$ ($\downarrow$) |
|---|---|---|
| HePCo (Ours) | $\mathbf{76.54 \pm 1.14}$ | $\mathbf{6.61 \pm 0.73}$ |
| Ablate previous server model | $61.15 \pm 2.13$ | $11.18 \pm 0.76$ |
| Ablate $\mathcal{L}_{KL}$ & $\mathcal{L}_{MSE}$ | $70.22 \pm 1.45$ | $8.86 \pm 0.66$ |
| Ablate $\mathcal{L}_{KL}$ | $71.39 \pm 1.34$ | $8.28 \pm 0.74$ |
| Ablate $\mathcal{L}_{MSE}$ | $74.11 \pm 1.31$ | $6.96 \pm 0.68$ |
| Ablate prompt distillation | $74.42 \pm 1.22$ | $7.02 \pm 0.88$ |
| Ablate classifier distillation | $68.46 \pm 0.91$ | $8.28 \pm 0.52$ |

Table 3: **Ablation Results (%) on 10-task CIFAR 100**. $A_N$ gives the accuracy averaged over tasks and $F_N$ gives the average forgetting.

## 6.1 Main Results

The results presented in Tables 1 and 2 demonstrate the dominant performance of our method in terms of average accuracy and forgetting across across all datasets and setups. The gains achieved by our method are more pronounced in the ImageNet-R setup which has longer task sequences and offer a significant shift from the pretrained distribution. Our approach achieves absolute improvements of up to 7% in average accuracy compared to the best baseline. All baselines that fine-tune the entire model are seen to struggle with longer sequences (CIFAR, Imagenet-R), showing significant forgetting. Under the class-balanced setting of Table 1, our approach achieves absolute improvements of more than **7%** on ImageNet-R in average accuracy compared to TARGET Zhang et al. (2023), which is the current SOTA. For the class-imbalanced settings in Table 2, our approach outperforms the competition by *even wider* margins. The notable performance drops observed across all methods highlight the complexity of this setting. Most importantly, HePCo achieves these solid results while enjoying low communication costs and without introducing any additional costs at the client-side. Furthermore, our approach faithfully aligns with the principles of Federated Learning (FL) by not assuming access to any storage, be it surrogate datasets or generated images, in contrast to methods like CFed Ma et al. (2022), TARGET Zhang et al. (2023) and GLFC Dong et al. (2022).

## 6.2 Additional Analysis

### 6.2.1 Ablation Studies

We perform ablations experiments on CIFAR-100 in the *No Imbalance* setting from Table 1.

**Ablating distillation of previous server model.** By removing the previous task server model from the distillation and generation steps, we highlight its importance in alleviating forgetting. By ablating this component, we observe a significant drop in performance indicated by a rise in forgetting ($F_N$) and a drop in average accuracy ($A_N$). The underlying intuition is that without the replay of past task data, the method strongly prioritizes learning of the current task leading to a loss of knowledge from previously seen tasks. In other words, using past task latents for replay mitigates inter-task forgetting.

**Ablating disagreement losses in generation.** To demonstrate the effectiveness of disagreement losses in generation, we set the lambda coefficients $\lambda_{KL}$ and $\lambda_{MSE}$ to zero and observe a 6% drop in accuracy. As discussed before, the intuition here is that in absence of the disagreement losses, the generator is prone to generate easily discriminable examples that lead to low classification loss but are less effective in distillation. To further highlight the importance of the individual losses, i.e $\mathcal{L}_{MSE}$ and $\mathcal{L}_{KL}$, we individually ablate them and observe performance drops.

**Ablating distillation sites.** Our approach uses pseudo-latents to fine-tune the key-prompt and classifier weights of the server model. In this experiment, we ablate the decision of fine-tuning the prompt components

| Method | Communication Cost ($\downarrow$) | Computation Overhead ($\downarrow$) | | Storage Overhead ($\downarrow$) | |
|---|---|---|---|---|---|
| | | Server-side | Client-side | Server-side | Client-side |
| CFed Ma et al. (2022) | 330.3 MB | 98s | 68s | 1433.6 MB | 1433.6 MB |
| TARGET Zhang et al. (2023) | 714.7 MB | 128s | 62s | 384.1 MB | 384.1 MB |
| **HePCo (Ours)** | 31.37 MB | 220s | N/A | 31.37 MB | N/A |

Table 4: Overhead costs incurred by our method against top performing baselines.

and the classifier separately and observe a decline in accuracy in both cases. The drop in performance is more pronounced when we do not perform distillation for the classifier. This experiment highlights our decision to fine-tune both prompt components and classifiers by operating in the latent space.

**Varying the category ratio.** Figure 3 shows the performance of all methods for different values of *category ratio*. We observe that HePCo consistently outperforms competing methods without requiring any hyperparameter or design changes. The performance gap between HePCo and the competing methods widens with decreasing category ratio, indicating its effectiveness in high heterogeneity settings.

### 6.2.2 Overhead Cost Analysis

**Memory & Communication Overhead.** Our method introduces additional parameters forming the prompting mechanism. The additional parameters amount to ~9.4% of the original size of the ViT encoder. Our method only needs to communicate the learnable parameters in the model which are the classifier and key-prompt components amounting to ~9.5% of the original model size. Methods that finetune the entire model need to learn and communicate all parameters in the encoder and classifier. Hence, our approach requires only 9.5% of the communication costs compared to all other methods that share complete models. Furthermore, the current state-of-the-art methods like CFed and TARGET require communicating a dataset of images (obtained from the surrogate dataset or a generative mechanism) after every round or task which significantly increases the communication overhead in addition to sharing complete models! For a ViT-B/16 architecture, sharing a complete model amounts to 330.3 MB of information. In addition to complete model weights, TARGET sends a buffer of 8k synthesized image-level data from server to clients to perform distillation leading to a communication volume of 714.7 MB. In contrast, our approach, only sends 31.37 MB of data in each client-server exchange. We report this information shared (in MB) during each client-server communication in Table 4.

**Computation Overhead.** Our method does not require any extra computation at the client side but introduces an overhead at the server side. This overhead includes the time required to train the generators and perform knowledge distillation. To quantify this overhead, we conducted benchmarking using 2 NVIDIA TITAN RTX GPUs in a 5 client setup, as described in the experiments section. We report results in Table 4 in terms of overhead time in seconds (s) per round. Our method adds an extra 220 seconds of computational time at the server side per round, in contrast to the 98 seconds introduced by CFed and the 128 seconds incurred by TARGET. It is crucial to emphasize that our method does not impose any additional overhead on the client side, unlike CFed and TARGET which incur 68 seconds and 62 seconds per client respectively. In those methods, the client is responsible for learning the current task as well as distilling knowledge from past tasks. By transferring the computation load from the client to the server, we prioritize client-level efficiency. In most practical federated learning scenarios, edge devices have limited computational capacity compared to the server. Our approach prioritizes client-level efficiency, even if it entails a slight trade-off in server-level efficiency.

**Storage Overhead.** As our method operates in a stateless FL setup, we do not require clients to maintain any state information or additional storage. Our approach requires the server model to store the classifier and prompt components corresponding to the last task model which is used in distillation resulting into a storage cost equal to ~9.5% of the base encoder model size. Other baselines Rebuffi et al. (2017a) incur extra storage costs at the client side equal to the size of entire encoder and classifier i.e ~86M parameters. Additionally, CFed and TARGET incur costs equivalent to storing an entire image dataset at both server and individual client levels. We report the storage costs incurred by these methods (in MB) at both server and client side in Table 4. The storage cost does not include the space needed to store the current round model itself. For

CFed, we use the Caltech-256 Griffin et al. (2007) as the surrogate dataset as prescribed in Ma et al. (2022). The storage overhead in Table 4 accounts for storing this dataset at both client and server sides.

*In summary, our approach attains state-of-the-art performance while imposing lower overheads compared to existing methods.*

## 7 Conclusion

In conclusion, we propose HePCo (Heterogeneous Prompt Consolidation) for continual federated learning. We formalize the setting and provide a methodical approach to simulate real-world conditions combining perspectives from continual and federated landscapes. Our method harnesses the prompt learning capabilities of foundation models to facilitate an efficient distillation framework for consolidating heterogeneous clients. By generating pseudo-data in a low-dimensional latent space, our approach enables parameter-efficient and data-free distillation of information from clients to server. We demonstrate the superior performance of our method compared to existing state-of-the-art methods through a series of experiments that emulate challenging real-world scenarios. By requiring clients to share parts of their models, we significantly reduce communication costs and enhance privacy. Importantly, our approach does not impose any additional overheads on the client side, making it highly valuable for real-world deployment.

## 8 Discussion

**Limitations.** It is worth noting that prompting-based methods are still relatively new and not extensively studied, making the explainability of these prompts challenging. Therefore, future work should focus on testing the robustness of these methods in diverse setups to ensure their effectiveness in different scenarios. Considering the typical asymmetry in the availability of computational resources across servers and clients, our approach prioritizes client-level efficiency. Yet, the computation overhead introduced at the server, may be an issue for some use-cases. Although the generation and distillation procedures are relatively lightweight, they still rely on availability of server-side compute resources, which may not be universally accessible in all scenarios. Additionally, our approach necessitates clients to use pretrained vision transformers, leaving open the question of how this framework can be extended to accommodate other architectures. These are interesting avenues for future research.

**Broader Impact.** The machine learning community is increasingly leaning towards the adoption of large-scale models for various applications. However, updating these models with new data poses a significant challenge. Retraining models from scratch each time new data arrives is computationally expensive and can have substantial financial Justus et al. (2018) and environmental Patterson et al. (2021); Lacoste et al. (2019) implications. Our approach offers a solution by enabling incremental learning on new data without the need for complete model retraining. Additionally, our use of prompting techniques allows for significant reductions in communication and local computation costs while enhancing privacy, which is especially critical for on-device edge computing applications.

## Acknowledgements

This material is based upon work supported by the National Science Foundation under Grant No. 2239292

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

## Appendix

## A  Algorithm

To better illustrate our proposed method, we present a whole picture of the method in Algorithm 1. The algorithm describes our complete procedure for a global task $T^i$, where $i \in [1, N]$.

## B  Experimental Details

**Implementation Details.** For fair comparison, we use the ViT-B/16 backbone pretrained on Imagenet-1K as the encoder for all methods. We implement our methods in PyTorch and use the PyTorch Image Models library Wightman (2019) to obtain pretrained checkpoints. We use 2 NVIDIA A40 GPUs for all experiments. For each result reported in this paper, we calculate the mean and standard deviation over separate runs.

**Training Details.** For all methods, we use the Adam Kingma & Ba (2017) optimizer with $\beta_1 = 0.9$ and $\beta_2 = 0.999$. We resize images to $224 \times 224$ and normalize to $[0,1]$.

**Hyperparameter Search.** Following DualPrompt Wang et al. (2022b), we use 20% of the training dataset as our validation data and conduct a hyperparameter search. We tune hyperparameters for both our approach and all competing baselines. We use a batch size of 64 for both local and server-side training, determined after searching over values of 16, 32, 64, 128. For our method and the prompting-based baselines, we use a learning rate of 1e-3, while for baselines that tune the entire model (FedAvg, FedLwF.MC), we use 5e-5. We search for learning rates among $\{1e^{-6}, 5e^{-5}, 1e^{-5}, 5e^{-4}, 1e^{-4}, 5e^{-3}, 1e^{-3}, 5e^{-2}, 1e^{-2}\}$. Our method employs a three-layer fully-connected network as the generator. We encode class labels using an embedding matrix of length 64 and concatenate it with a 64-dimensional noise vector. This dimension was chosen after searching over 32, 64, 128, 256. While our approach is robust to various values, 64 performs best. The generator architecture has input sizes of [128, 256, 1024] per layer, with an output size of 768 which is the dimension of the visual query. We opt for a three-layer architecture to maintain lightweight generation and training. We train the generator for 100 epochs using a batch size of 64 and a learning rate of $1e^{-4}$ using the Adam optimizer. After a logarithmic scale search, we find a learning rate in the range [5e-5, 1e-4] to provide optimal results.

We fine-tune the server model using a learning rate of $1e^{-4}$ for 200 epochs. Through grid search, we find that the model is robust to various learning rate and epoch combinations, with these values providing the best average accuracy on the validation set. We use a *replay ratio* of 0.5 for our method, which means we mix 50 pseudo-latents corresponding to previous tasks for every 100 pseudo-latents corresponding to the current task. We conduct a search over values like [0, 0.125, 0.25, 0.375, 0.5, 0.625, 0.75, 0.875, 1] and find 0.5 to result into the best average accuracy $A_N$. We observe a *stability-plasticity* trade-off controlled by this hyperparameter with larger values leading to lower forgetting ($F_N$) but lower current task accuracies (*plasticity*) and smaller values yielding the opposite effect. Through a hyperparameter search within [0, 5] at 0.1 increments, we choose $\lambda_{KL}$ and $\lambda_{MSE}$ values to be 1 and 0.1 respectively. We find these values to work best across all datasets reported in the paper. Overall, $\lambda_{KL}$ values between [0.6, 3] yield similarly good results, with too low (close to 0) and too high (close to 5) values leading to poor performance.

---

**Algorithm 1** HePCo: Heterogeneous Prompt Consolidation

---

**Task**: For each task $T^i$, the server is trained through $R$ communication rounds
**Input**: Set of $\mathcal{C}$ clients with trainable parameters $\mathbf{w}_c = \{P_c, K_c\}, \phi_c$, pretrained ViT parameters $\theta_{pt}$, trainable server parameters $\mathbf{w}_s = \{P_s, K_s\}, \phi_s$, local epochs $E$, generator training epochs $E_g$, distillation epochs $E_d$
**Output**: $\mathbf{w_s}, \phi_s$

1: **Server executes**:
2: Send the pretrained model parameters $\theta_{pt}$ to clients.
3: **for** $r \in \{1, \cdots, R\}$ **do**
4:    **for** each client $c \in \mathcal{C}$ **do**
5:       $\mathbf{w}_c \leftarrow \mathbf{w}_s$
6:       $\phi_c \leftarrow \phi_s$
7:       $\mathbf{w_c}, \phi_c \leftarrow$ **Client Update**$(\mathbf{w_c}, \phi_\mathbf{c})$
8:    **end for**
9:    $\theta_\mathbf{gen} \leftarrow$ **Train Generator**
10:   Obtain provisional server model weights $\mathbf{w_s}$ and $\phi_\mathbf{s}$ by averaging client weights.
11:   $\mathbf{w_s}, \phi_\mathbf{s} \leftarrow$ **Latent Distillation**$(\mathbf{w_s}, \phi_\mathbf{s}, \theta_\mathbf{gen})$
12: **end for**

1: **Train Generator**:
2: **for** $e \in E_g$ **do**
3:    $\mathcal{L}_{cls} \leftarrow$ Calculate *Cross Entropy loss* in equation 2
4:    $\mathcal{L}_{KL} \leftarrow$ Calculate *Kullback-Leibler divergence* in equation 4a
5:    $\mathcal{L}_{MSE} \leftarrow$ Calculate *Mean-Squared Error* in equation 4b
6:    Update $\theta_\mathbf{gen}$ using equation 5
7: **end for**
8: **return** $\theta_\mathbf{gen}$

1: **Latent Distillation** $(\mathbf{w_s}, \phi_\mathbf{s}, \theta_\mathbf{gen})$:
2: **for** $e \in E_d$ **do**
3:    $\epsilon \sim \mathcal{N}(\mathbf{0}, \mathbf{1})$
4:    Sample class labels $y$ randomly from current and past task categories
5:    Obtain pseudo-latents $z = G(\epsilon, y; \theta_{gen})$ from generator
6:    Obtain distillation target prompts from clients and previous server model
7:    $\mathcal{L}_{prompt} \leftarrow$ Calculate *Mean-Squared Error* in equation 6
8:    Update $\mathbf{w}_s$
9:    Calculate *cross-entropy loss* between sampled class labels $y$ and predictions $\phi_s(z)$
10:   Update $\phi_\mathbf{s}$
11: **end for**
12: **return** $\mathbf{w}_s, \phi_s$

1: **Client Update** $(\mathbf{w_c}, \phi)$:
2: Freeze ViT backbone $\theta_{pt}$
3: **for** $e \in E$ **do**
4:    Calculate visual query feature $q(x)$ for image $x$
5:    Obtain final prompt $p$ from $q(x)$ using $P_c$ and $K_c$
6:    Insert $p$ into selected layers of ViT $\theta_{pt}$
7:    $\mathcal{L}_{CE} \leftarrow$ Calculate *cross entropy* loss $\mathcal{L}_{CE}$
8:    Update $\mathbf{w}_c, \phi_c$
9: **end for**
10: **return** $\mathbf{w}_c, \phi_c$

---

