# OpenReview forum: "Continual Adaptation of Vision Transformers for Federated Learning"
_TMLR — Accepted by TMLR_

### Review · Reviewer_aaiv · 2024-07-09

**Summary Of Contributions:**

This paper studies continual learning in the setting of federated learning. On the client side, it uses a frozen vision encoder and learns prompts and classifiers. On the server side, it reconstructs images in the latent space and fine-tunes the prompts and classifier according to the latent images. The proposed method is shown to achieve higher accuracy than existing methods.

**Audience:**

Yes

**Broader Impact Concerns:**

I do not see any ethical concerns for this paper.

**Claims And Evidence:**

Yes

**Requested Changes:**

Adress all comments in the weakness part.

R1: Clarify the privacy benefits of the proposed method over existing methods.

R2: Enhance the efficiency analysis with concrete experiment results.

R3: Clarify how existing researches work such that readers can understand their limitations.

R4: Discuss whether and how the proposed method can work with other types of data.

**Strengths And Weaknesses:**

Strength

S1: Transferring the client prompts and classifiers is more efficient than transferring the client models.

S2: Generating pseudo images in the latent space is cheaper than generating the original images.

S3: The experiment results show that the proposed method achieves good accuracy.


Weakness

W1: It is unclear why the proposed method achieves better data privacy than methods that transfer client models. I understand that if the server has client models, it can use model inversion techniques to reconstruct the original images. In the proposed method, the server can also use model inversion if it knows the frozen vision encoder. This is necessary because the server needs to know the vision encoder for model serving.

W2: The efficiency advantage is analyzed over many places but not properly established. It will be better if the authors can provide concrete numbers (e.g., via a table) for the communication and computation volume of the clients in the training process.

W3: The discussions of related work can be enhanced, especially the continual federated learning part. It is difficult to understand why existing methods are not efficient from the current discussions. For instance, the authors say “FedWeiT Yoon et al. (2021) aims to learn better client models by minimizing interference across client weights. FedWeiT incurs considerable overheads in terms of communication, computation and storage.” However, where do these overheads come from? The authors should describe how FedWeiT works in addition to the high-level ideas.

W4: The method and experiment parts consider image datasets. The paper can be enhanced if the authors can discuss how the proposed methods may be used for other types of data (e.g., tabular).

---

> ### Author Response · Authors · 2024-08-04
> **Author Response to Reviewer aaiv (Part 1/2)**
>
> We thank the reviewer for the insightful feedback on our paper. We address the reviewer comments below:
>
> > It is unclear why the proposed method achieves better data privacy than methods that transfer client models. I understand that if the server has client models, it can use model inversion techniques to reconstruct the original images. In the proposed method, the server can also use model inversion if it knows the frozen vision encoder. This is necessary because the server needs to know the vision encoder for model serving.
>
> In our approach, the server knowing the vision encoder is not sufficient to reconstruct individual client models. Reconstruction would require information about where to insert specific client-trained prompts—information the server lacks. Specifically, our approach employs a prompting based approach called *decomposed prompting* at the client side; which is built on top of L2P. Like in L2P, prompts are inserted into a subset of the layers of a ViT. This subset of layers is a hyperparameter determined by the engineer. The exact locations (layers) of inserting these prompts are known only to clients. Our approach requires clients to share only the prompt and classifier weights, without such knowledge of where insertion happens. To perform generation and distillation, it is not required to reconstruct entire client models, and merely using the prompt weights (as is) suffices. The server therefore lacks knowledge of the exact locations for inserting the prompts, as this information is known only to the respective clients. If the server tried to reconstruct the exact client model, it would need to conduct an exhaustive search over the number of layers in the transformer and try various combinations to determine the precise insertion positions. In this manner, our method represents a positive step towards enhancing client privacy compared to approaches that share complete model weights. Nonetheless, we slightly temper our privacy claims in the paper (p2: “ensur[e] client privacy -> improv[e] client privacy”) as it is difficult to verify the privacy guarantees of this approach.
>
> > The efficiency advantage is analyzed over many places but not properly established. It will be better if the authors can provide concrete numbers (e.g., via a table) for the communication and computation volume of the clients in the training process.
>
> Thank you for the suggestion. We have added a new table in the Section 6.2.2 (Table 4) to quantify communication, computation and storage overhead incurred by the methods. We also elaborate on the metrics and methods used to compute these overhead costs in Section 6.2.2.
>
> > The discussions of related work can be enhanced, especially the continual federated learning part. It is difficult to understand why existing methods are not efficient from the current discussions. For instance, the authors say “FedWeiT Yoon et al. (2021) aims to learn better client models by minimizing interference across client weights. FedWeiT incurs considerable overheads in terms of communication, computation and storage.” However, where do these overheads come from? The authors should describe how FedWeiT works in addition to the high-level ideas.
>
> Following this suggestion, we have updated the related work section to elaborate the working of these methods and shed further light on their limitations. Additionally, In response to reviewer TAtt’s comments, we have included two recent baselines to this discussion: Fed-CPrompt (Bagwe et. al. 2023) and GAL (Wang et. al. 2023). Furthermore, we report results for Fed-CPrompt for all datasets and benchmarks in the paper. We observe that our proposed method, HePCo, noticeably outperforms this baseline across the board. In Table 4 and Section 6.2.2, we discuss in detail the efficiency benefits offered by our approach in comparison to state-of-the-art baselines.

---

> ### Author Response · Authors · 2024-08-04
> **Author Response to Reviewer aaiv (Part 2/2)**
>
> > The method and experiment parts consider image datasets. The paper can be enhanced if the authors can discuss how the proposed methods may be used for other types of data (e.g., tabular).
>
> The prompting scheme described in our paper inherits from popular approaches in continual learning: L2P (Wang et. al 2022) and CODA-Prompt (Smith et.al 2023). As these foundational works have solely focused on image datasets, our current implementation is similarly constrained to image classification tasks. We have updated the title of our paper to better reflect this scope. We agree that using this method for other domains would be a great area of future work.
>
> References:
> 1. Gaurav Bagwe, Xiaoyong Yuan, Miao Pan, & Lan Zhang. (2023). Fed-CPrompt: Contrastive Prompt for Rehearsal-Free Federated Continual Learning.
> 2. Lixu Wang, Chenxi Liu, Junfeng Guo, Jiahua Dong, Xiao Wang, Heng Huang, & Qi Zhu. (2023). Federated Continual Novel Class Learning.
> 3. Zifeng Wang, Zizhao Zhang, Chen-Yu Lee, Han Zhang, Ruoxi Sun, Xiaoqi Ren, Guolong Su, Vincent Perot, Jennifer Dy, & Tomas Pfister. (2022). Learning to Prompt for Continual Learning.
> 4. James Seale Smith, Leonid Karlinsky, Vyshnavi Gutta, Paola Cascante-Bonilla, Donghyun Kim, Assaf Arbelle, Rameswar Panda, Rogerio Feris, & Zsolt Kira. (2023). CODA-Prompt: COntinual Decomposed Attention-based Prompting for Rehearsal-Free Continual Learning.

---

> > ### Comment · Reviewer_aaiv · 2024-08-14
> > **Thank for the response, my concerns are addressed**
> >
> > Thank for the response, my concerns are addressed

---

### Review · Reviewer_TAtt · 2024-07-15

**Summary Of Contributions:**

The paper looks at the problem of continual federated learning (CFL) in a class-incremental cross-device setting, i.e., with many stateless clients and with new classes appearing with new tasks in a sequential fashion. The proposed method is based on client-side prompt-learning, that closely builds on existing CL methods, and a novel server-side knowledge distillation approach to aggregate the heterogeneous client-contributions. The authors empirically show that their approach can improve on existing baselines in several settings for image recognition.

**Audience:**

Yes

**Broader Impact Concerns:**

I have no broader impact concerns for this paper.

**Claims And Evidence:**

No

**Requested Changes:**

In decreasing order of importance:

1) It is hard for me to get the exact flow of what is happening on the clients and on the server in the proposed method and how exactly these link together, even given Figs 1 & 2. I would propose you add explicit server and client side algorithms to make this clearer.

2) It is generally a bit hard to see which parts are meant as novel contributions and which are simply restating background information from existing methods, as the background is covered here and there in Sec4 while introducing the proposed method. I would propose that you clearly separate the background from the proposed method on the level of sections to make this clear.

3) Please explain why you do not include some existing methods in any of the comparison and discussion (FedWeiT, FedCIL mentioned in Sec2 related work, GAL from Wang et al. 2023, Fed-CPromp from Bagwe et al. 2023; also add Wang et al. and Bagwe et al. to the discussion about existing work).

4) In several places (see pages 1-5, 7, 11) you claim in so many ways that the presented method is privacy-preserving, yet there is no actual definition of privacy given in the paper, either formally or informally. While some of the claims make sense on an intuitive level, e.g., not requiring a server-side replay buffer holding actual client data is better than requiring it, in many places the method is claimed to be privacy-preserving without comparing it to such an obvious baseline (e.g. p2: goal is to "ensur[e] client privacy"; I do not see how the proposed method would guarantee privacy in any meaningful sense). I find such claims unsupported, as communicating fewer parameters does not necessarily mean that the method is any more private than a baseline sending more parameters, or that the proposed method could not leak client data in the clear to an adversary.

5) Based on the title and abstract I would assume you look at different foundation models (FM) or at least present a method which is more or less directly applicable to any FM. However, the actual method and all the experiments look solely at visual transformer (ViT) models, and seem to rely on having attention layers in the model. I would propose you change the title and abstract to better fit the actual work described in the main paper.

6) In several places there hard-to-verify claims (e.g. , achieving high accuracy in describing generator on p6, efficiently fine-tuning at the start of Sec4.4). Please clarify in which sense these claims are true, and point to the supporting evidence in the text.

7) How sensitive the results are to the hyperparameter tuning? Please state (at least in the appendix) if you have done some other hyperparameter search (e.g. as currently stated for lr), and whether you have also tuned all the baseline hyperparams.

8) Please include some deviation measure in Fig3.

9) In the results (e.g. Tables 1 & 2), consider (half-)highlighting methods that overlap with the best mean given the reported deviation.

10) abstract: "understudied problem": there are plenty of papers on the topic, so how exactly is this understudied?

11) p1: claiming that standard FL assumes iid data is misleading; as you state next, the problem of heterogeneous data is a well-studied in existing FL literature. Please refine this claim.

12) p2, related to contributions: please also state how you measure communication costs (eg, number of parameters sent vs number of communication rounds) and computational efficiency when claiming that the proposed method is efficient in these respects.

13) p4: "Formalizing the setting in this manner enables us to methodically vary the parameters, thereby simulating a heterogeneous setting that closely resembles the complexities encountered in the real world": do you actually have some real-world data or settings that the data splits you use closely mimic? If not, please do not claim close resemblance.

14) Table 3: $F_N$ is not included in the results although mentioned in the caption and in the text.

### References:

Bagwe et al. 2023: Fed-CPrompt: Contrastive Prompt for Rehearsal-Free Federated Continual Learning.

Wang et al. 2023: Federated Continual Novel Class Learning.

**Strengths And Weaknesses:**

### Strengths
* By and large, the proposed method seems to make sense.
* The authors present some empirical evidence that the proposed method improves on the existing baselines.
* The paper includes some ablation results pointing to the relative importance of the various parts of the proposed method.

### Weaknesses
* The paper is a bit hard to follow, and the writing could be improved.
* The experiments do not seem to include all existing methods as baselines.
* Some of the claims are either unsupported or their status is hard to assess.

---

> ### Author Response · Authors · 2024-08-04
> **Author Response to Reviewer TAtt (Part 1/3)**
>
> We sincerely thank the reviewer for their thorough and constructive feedback. We address the reviewer comments below:
>
> > It is hard for me to get the exact flow of what is happening on the clients and on the server in the proposed method and how exactly these link together, even given Figs 1 & 2. I would propose you add explicit server and client side algorithms to make this clearer.
>
> Thank you for the comment. Making this more explicit and detailed is a great idea, and in response we have added server and client algorithms to detail our method in Appendix section A.
>
> > It is generally a bit hard to see which parts are meant as novel contributions and which are simply restating background information from existing methods, as the background is covered here and there in Sec4 while introducing the proposed method. I would propose that you clearly separate the background from the proposed method on the level of sections to make this clear.
>
> We thank the reviewer for this suggestion. We have separated the background from our novel contributions on the level of sections as suggested.
>
> > Please explain why you do not include some existing methods in any of the comparison and discussion (FedWeiT, FedCIL mentioned in Sec2 related work, GAL from Wang et al. 2023, Fed-CPromp from Bagwe et al. 2023; also add Wang et al. and Bagwe et al. to the discussion about existing work).
>
> Following the reviewer’s suggestion, we have now included Fed-CPrompt as a baseline for comparison across all of our benchmarks. We have also added a discussion to provide context about this method and highlight the differences against our proposed approach. Our approach, HePCo, is shown to outperform this method across all datasets and benchmarks reported in the paper. In reference to the other baselines suggested by the reviewer, we now provide further context about FedWeiT and FedCIL in our related works section, and have newly added GAL to it. Comparing these methods is non-trivial; particularly, the aim of FedWeiT is to train multiple continual learning agents (client models) that benefit from each other’s indirect experience. In contrast, we are focused on developing a single global model (server) that is built from client models trained on local data. We choose not to compare to FedCIL as its method necessarily requires an ACGAN (Odena et. al 2017) architecture which prevents fair comparison with our Vision Transformer (ViT) based models. Finally, GAL introduces a new task of Federated Continual Novel Class Learning (FedCN) which is completely different from our problem of federated class incremental learning. Particularly, FedCN operates in an unlabelled setting, where methods are expected to discover and learn unlabelled novel class data. In contrast, our work explores a supervised continual learning setup where new labeled instances are incrementally introduced.

---

> ### Author Response · Authors · 2024-08-04
> **Author Response to Reviewer TAtt (Part 2/3)**
>
> >  In several places (see pages 1-5, 7, 11) you claim in so many ways that the presented method is privacy-preserving, yet there is no actual definition of privacy given in the paper, either formally or informally. While some of the claims make sense on an intuitive level, e.g., not requiring a server-side replay buffer holding actual client data is better than requiring it, in many places the method is claimed to be privacy-preserving without comparing it to such an obvious baseline (e.g. p2: goal is to "ensur[e] client privacy"; I do not see how the proposed method would guarantee privacy in any meaningful sense). I find such claims unsupported, as communicating fewer parameters does not necessarily mean that the method is any more private than a baseline sending more parameters, or that the proposed method could not leak client data in the clear to an adversary.
>
> Our claim is that our method improves privacy (not guarantee it, as that would be difficult), not just because a subset of parameters are sent but also because one would have to search over a large search space in terms of where to use such parameters (before inversion). Specifically, our approach employs a prompting based approach called *decomposed prompting* at the client side, which is built on top of L2P. Like in L2P, prompts are inserted into a subset of the layers of a ViT. This subset of layers is a hyperparameter determined by the engineer. The exact locations (layers) of inserting these prompts are known only to clients. Our approach requires clients to share only the prompt and classifier weights, without such knowledge of where insertion happens. To perform generation and distillation, it is not required to reconstruct entire client models, and merely using the prompt weights (as is) suffices. The server therefore lacks knowledge of the exact locations for inserting the prompts, as this information is known only to the respective clients. If the server tried to reconstruct the exact client model, it would need to conduct an exhaustive search over the number of layers in the transformer and try various combinations to determine the precise insertion positions. In this manner, our method represents a positive step towards enhancing client privacy compared to approaches that share complete model weights. Nonetheless, we slightly temper our privacy claims in the paper (p2: “ensur[e] client privacy -> improv[e] client privacy”) as indeed it is difficult to verify the privacy guarantees of this approach.
>
> > Based on the title and abstract I would assume you look at different foundation models (FM) or at least present a method which is more or less directly applicable to any FM. However, the actual method and all the experiments look solely at visual transformer (ViT) models, and seem to rely on having attention layers in the model. I would propose you change the title and abstract to better fit the actual work described in the main paper.
>
> Thank you for raising this issue; we agree and have modified the title and abstract. Our method relies on existing works that explore prompting for continual learning: L2P (Wang et. al 2022), CODA-Prompt (Smith et.al 2023) etc. that have restricted their studies to vision transformer backbones. We have updated our title to “Continual Adaptation of Vision Transformers for Federated Learning” to better represent our scope.
>
> > In several places there hard-to-verify claims (e.g. , achieving high accuracy in describing generator on p6, efficiently fine-tuning at the start of Sec4.4). Please clarify in which sense these claims are true, and point to the supporting evidence in the text.
>
> We agree and have modified the text accordingly. We have rephrased the line on p6: “helps combat intra-task forgetting and achieves a high accuracy on the global data distribution…” -> “helps combat intra-task forgetting and allows the server model to achieve a high accuracy on the global data distribution...” This can be verified from the average accuracy metric ($A_N$) from Table 1 and our ablation results in Table 3.
>
> With regard to the “efficient fine-tuning” claim in Sec 4.4 (which is now Section 5.3 in the updated paper), we refer to the fact that we only need to tune specific layers of the model (only prompt and classifier weights) without needing a forward pass through the whole model. This process is efficient in comparison to typical distillation setups, where all parameters are updated through forward and backward passes through the entire model.

---

> ### Author Response · Authors · 2024-08-04
> **Author Response to Reviewer TAtt (Part 3/3)**
>
> > How sensitive the results are to the hyperparameter tuning? Please state (at least in the appendix) if you have done some other hyperparameter search (e.g. as currently stated for lr), and whether you have also tuned all the baseline hyperparams.
>
> We have updated the Appendix to include a detailed discussion on how we choose all hyperparameters from our method. We find our method to be robust to several choices of hyperparameters. In Appendix Section B, we provide numerical ranges for hyperparameters within which our method consistently performs well.
>
> > In the results (e.g. Tables 1 & 2), consider (half-)highlighting methods that overlap with the best mean given the reported deviation.
>
> Thank you. We have incorporated this suggestion into our updated version.
>
> >  abstract: "understudied problem": there are plenty of papers on the topic, so how exactly is this understudied?
>
> This is a relatively new area, with almost every work proposing new task definitions, benchmarks and methodologies. Further, many of these papers have concurrently come out recently (around the time that this work was done). As the community hasn’t yet settled on a benchmark or task setup yet, we call this an understudied problem.
>
> > Suggestions [11-13]
>
> Thank you. We have refined the text by incorporating these suggestions.
>
> > Table 3: FN is not included in the results although mentioned in the caption and in the text.
>
> Thank you for pointing this out. We now include forgetting results for this table as well.
>
> References:
> 1. Augustus Odena, Christopher Olah, and Jonathon Shlens. Conditional image synthesis with auxiiliary classifier gans. In International conference on machine learning, pp. 2642–2651. PMLR, 2017.
> 2. Zifeng Wang, Zizhao Zhang, Chen-Yu Lee, Han Zhang, Ruoxi Sun, Xiaoqi Ren, Guolong Su, Vincent Perot, Jennifer Dy, & Tomas Pfister. (2022). Learning to Prompt for Continual Learning.
> 3. James Seale Smith, Leonid Karlinsky, Vyshnavi Gutta, Paola Cascante-Bonilla, Donghyun Kim, Assaf Arbelle, Rameswar Panda, Rogerio Feris, & Zsolt Kira. (2023). CODA-Prompt: COntinual Decomposed Attention-based Prompting for Rehearsal-Free Continual Learning.

---

> > ### Comment · Reviewer_TAtt · 2024-08-06
> > **No further questions**
> >
> > Thanks for the thorough update, I have no further questions on this paper.

---

### Review · Reviewer_iEXM · 2024-07-21

**Summary Of Contributions:**

This paper focuses on the continual federated learning problem, where a server communicates with a set of clients to incrementally learn new concepts over time without sharing or storing any data. The authors study this problem in the context of Foundation Models and explore parameter-efficient approaches to adapt to dynamic distributions while minimizing forgetting. They propose to leverage prompting-based methods and distill data from both the past task label distribution as well as the current task label distribution. Numerically, their proposed approach outperforms both existing methods and contributed baselines by as much as 7\% while reducing communication costs by only sharing a small set of model parameters. The paper is in general interesting and well-written, but I have the following comments.

**Audience:**

Yes

**Broader Impact Concerns:**

The robustness of the prompting-based methods need to be tested across diverse setups to ensure their effectiveness in different scenarios.

**Claims And Evidence:**

Yes

**Requested Changes:**

Comments:

- Definition of $L_{cls}^c$: $\mathcal{L}_{cls}^c$ is interpreted as the cross-entropy loss between the prediction of local model $c$ given latent $z$ and sampled class label $y$. It is defined for the local model $c$. However, in the right-hand-side of Equation (3), it sums over all local models in $C$. Please define precisely.

- What is $\mathcal{L}_{dis}$ defined on page 7 line 5?

- In Equation (6), why choosing $\zeta_{t-1}^y$ as an indicator variable instead of a weighted indicator variable? That is, $L_{prompt}= \sum_{c\in C} L_{MSE}^c + w*\zeta_{t-1}^y L_{MSE}^{t-1}$ where $w$ is a parameter?

- For the ablation study, could you also report the metric of average forgetting?

- In Table 2, $\beta$, which is an imbalance ratio, is chosen as 0.05 and 0.01. Could you report the experiment results for a wider range of $\beta$ (e.g., 0 to 1)?

- How do you choose $\lambda_{KL}$ and $\lambda_{MSE}$ in the numerical experiments on different datasets?

Minor comments:

-  The sentence below (3) "where... and sampled class label y": y should be in math mode $y$.

- The sentence before Equation (5) "for these these losses": delete "these"

**Strengths And Weaknesses:**

Strengths: The paper is clearly written and the proposed method shows a strong numerical performance.

Weaknesses: See my comments below.

---

> ### Author Response · Authors · 2024-08-04
> **Author Response to Reviewer iEXM**
>
> We thank the reviewer for their constructive feedback on our paper. We address the reviewer comments below:
>
> > Definition of $L^c_{cls}:L^c_{cls}$ is interpreted as the cross-entropy loss between the prediction of local model $c$ given latent $z$ and sampled class label $y$. It is defined for the local model $c$. However, in the right-hand-side of Equation (3), it sums over all local models in $C$. Please define precisely.
>
> Thank you. The total classification loss is calculated as the sum of cross-entropy loss for each client. For each client, the class label $y$ is randomly sampled from the categories encountered by the client.
>
> > What is $L_{dis}$ defined on page 7 line 5?
>
> Thank you for bringing this to our attention. It was a typographical error that we have corrected now by replacing $L_{dis}$ by $L_{KL}$ and $L_{MSE}$ in the paper.
>
> > In Equation (6), why choosing $\zeta_{t-1}^{y}$ as an indicator variable instead of a weighted indicator variable? That is, $L_{prompt} = \sum_{c\in C}L_{MSE}^c + w*\zeta_{t-1}^{y}{L}_{MSE}^{t-1} $ where $w$ is a parameter?
>
> This is an interesting point. Based on our current results, we find that such parametric flexibility is not necessary, as evidenced by our results. Also, introducing a weighted indicator as a hyperparameter would necessitate additional tuning. We've deliberately chosen to maintain simplicity by keeping this as a binary variable, thereby avoiding the need for further hyperparameter optimization.
>
> > For the ablation study, could you also report the metric of average forgetting?
>
> Thank you for pointing this out. We have updated the paper to include the results for average forgetting in Table 3.
>
> > How do you choose $\lambda_{KL}$ and $\lambda_{MSE}$ in the numerical experiments on different datasets?
>
> We determine values for $\lambda_{KL}$ and $\lambda_{MSE}$ through hyperparameter search on hold-out validation sets for each dataset. We find $\lambda_{KL}$ = 1 and $\lambda_{MSE}$ = 0.1 to work best across all datasets and configurations. Overall, $\lambda_{KL}$ values between [0.6, 3] yield similarly good results, with too low (close to 0) and too high (close to 5) values leading to poor performance.
>
> > Minor comments…
>
> Thank you. We have addressed these comments in the updated paper.

---

### Author Response · Authors · 2024-08-04
**Update from Authors - Revision Uploaded**

We thank all the reviewers for their thorough and constructive feedback. We believe we have addressed all reviewer requests in our revised submission (revisions are marked in our document with blue text). Additionally, we have provided point-by-point responses to all reviewers to address their concerns. Here we summarize the main modifications of our paper:

- We have added explicit client and server algorithms to further illustrate our method (Reviewer TAtt; Appendix Section A)
- We have included a new baseline: Fed-CPrompt (Bagwe et. al. 2023) and reported results for all benchmarks. (Reviewer TAtt; Table 1 and 2). Our proposed method, HePCo, noticeably outperforms this baseline across all benchmarks and metrics.
- We have further elaborated on the related works, focusing on their methodology and highlighting differences from our approach. Additionally, we include a new related work: GAL (Wang et. al. 2023) in this section. (Reviewers TAtt, aaiv; Section 2)
- We have clarified the privacy-improving aspects of our method and have adjusted relevant text in the paper. (Reviewers TAtt, aaiv; Section 5.1)
- We have updated the title and abstract of our paper to better reflect the scope of our method. (Reviewers TAtt, aaiv)
- We have discussed hyperparameter selection strategies in greater detail for all hyperparameters involved in our approach (Reviewers TAtt, iEXM, aaiv; Appendix)
- We have provided concrete results for analyzing the efficiency of our method to better support our claims. These results highlight the efficiency benefits that our approach offers over existing methods. (Reviewers aaiv, TAtt)
- We have included additional metrics for existing experiments as requested. (Reviewers TAtt, iEXM, aaiv)
- We have addressed all writing level suggestions by editing the corresponding text in the paper. (Reviewers iEXM, TAtt, aaiv)

We believe these improvements have significantly improved the paper and thank the reviewers for their constructive feedback!

---

### Decision · Action_Editor_wMje · 2024-09-02

**Recommendation:** Accept as is

**Comment:**

Based on the overall positive reviews, the authors' comprehensive response to the reviewers' comments, and the technical merits of the proposed method, I recommend accepting this paper. The authors have well addressed the major weaknesses and provided a clear and detailed presentation of their work.

**Audience:**

yes

**Claims And Evidence:**

yes